# The Prevalence and Correlates of Labor and Sex Trafficking in a Community Sample of Youth Experiencing Homelessness in Metro-Atlanta

Eric R. Wright [1,2,*], Ana LaBoy [1,3], Kara Tsukerman [1], Nicholas Forge [4], Erin Ruel [1], Renee Shelby [5], Madison Higbee [1], Zoe Webb [4], Melanie Turner-Harper [1], Asantewaa Darkwa [1] and Cody Wallace [1]

1   Department of Sociology, Georgia State University, Atlanta, GA 30302, USA; alaboy1@gsu.edu (A.L.); kkoplan1@gsu.edu (K.T.); eruel@gsu.edu (E.R.); mhigbee1@student.gsu.edu (M.H.); mturner19@gsu.edu (M.T.-H.); adarkwa1@gsu.edu (A.D.); cwallace26@student.gsu.edu (C.W.)
2   School of Public Health, Georgia State University, Atlanta, GA 30302, USA
3   Georgia Health Policy Center, Georgia State University, Atlanta, GA 30302, USA
4   School of Social Work, Georgia State University, Atlanta, GA 30302, USA; nforge1@gsu.edu (N.F.); zwebbfc@gmail.com (Z.W.)
5   Program in Gender and Sexuality, Northwestern University, Evanston, IL 60208, USA; renee.shelby@northwestern.edu
*   Correspondence: ewright28@gsu.edu; Tel.: +1-404-413-6527

**Abstract:** Research suggests that runaway and homeless youth (RHY) in the United States are vulnerable to sex and labor trafficking. In this paper, we report and analyze estimates of sex and labor trafficking collected as part of the Atlanta Youth Count 2018, a community-based field survey of RHY between the ages of 14 and 25 in the metro-Atlanta area. A total of 564 participants were recruited and completed a survey that included questions about their backgrounds as well as the Human Trafficking Screening Tool (HTST). We found that 39.9% experienced some form of trafficking while homeless. While 15.6% of the youth reported commercial sexual exploitation while homeless, coerced labor (29.3%) or fraud (25.2%) were even more common experiences. Women, transgender, and gender nonconforming youth, as well young people who had prior system involvement and those who had been homeless for more than a year were the most likely to report having been trafficked. The significance of these findings for research and policy on RHY and trafficking are discussed.

**Keywords:** youth; young adults; homeless; homelessness; human trafficking; sex trafficking; labor trafficking

## 1. Introduction

A growing body of research suggests that runaway and homeless youth (RHY) in the United States are vulnerable to sex and labor trafficking. Long recognized as a vulnerable population (Institute of Medicine and National Research Council 2013), our understanding of the prevalence and characteristics of RHY who experience trafficking is limited because these young people are an elusive, mobile, and generally hidden population. To date, studies have relied predominantly on data collected from youth in shelter settings. Furthermore, their approach to measuring trafficking has varied and been primarily focused on minors' involvement in commercial sexual exploitation (Gibbons et al. 2020; Institute of Medicine and National Research Council 2013). While these studies have been critical in raising awareness of this important social problem, more research is needed to understand the nature and extent of exploitative experiences in order to build effective and victim-centered institutional support services and policies that address their needs.

In this paper, we report estimates of commercial sexual exploitation, labor, and any trafficking collected during the Atlanta Youth Count 2018 field study (AYC18). In contrast to past shelter-based studies, the AYC18 study involved the collection of a community-wide

sample of RHY across the metro-Atlanta area, including young people recruited in shelters, on the streets, in motels/hotels, as well as other public locations where homeless people congregate. In addition, this study adopted the recently validated the Human Trafficking Screen Tool (HTST; Dank et al. 2017) to better capture a broader range of trafficking experiences RHY may encounter. In addition to estimating the overall prevalence of trafficking, we breakout trafficking by various demographic and social characteristics associated with a greater risk of sex, labor, and any trafficking while homeless.

*1.1. Background*

Human trafficking is the "acquisition of people by improper means such as force, fraud or deception, with the aim of exploiting them" for sexual acts or labor services (United Nations Office on Drugs and Crime 2014). In 2000, the United States Victims of Trafficking and Violence Protection Act made trafficking a federal crime. Trafficking is understood to be one of the most lucrative underground economic activities, with thousands of annual victims, including commercially sexually exploited children (CSEC), adults forced or coerced into providing paid sexual services, and individuals regardless of age forced into other forms of labor or services (Farrell et al. 2012). Over 63,300 cases of human trafficking have been handled by the U.S. National Human Trafficking Hotline since 2007. In 2019, the hotline received 11,500 reported cases involving 22,326 victims, up almost 20% from 2018. Among the victims, 14,597 were sex trafficking, 4934 labor trafficking, 1048 reported both types, and 1747 were not specified. The reported cases include over 5300 minors and 15,532 people who did not give their age (National Human Trafficking Hotline 2020).

Runaway and homeless youth (RHY) are a vulnerable population. They often report past or current involvement in the child welfare system, substance use, and mental illness (Davies and Allen 2017; Edidin et al. 2012; Gambon and O'Brien 2020; Dank et al. 2015; Institute of Medicine and National Research Council 2013; Narendorf 2017). RHY also have been found to be disproportionately likely to be lesbian, gay, bisexual, transgender, or queer or questioning (LGBTQ), having run away or been kicked out by disapproving caretakers (Forge et al. 2018; Gambon and O'Brien 2020; Kattari and Begun 2017; Keuroghlian et al. 2014; Maccio and Ferguson 2016; Ream and Forge 2014; Dank et al. 2015). Long considered a "hidden" population, RHY are more mobile and more likely than adults experiencing homelessness to sleep outside or in precarious doubled-up situations (Hallett 2012; Henry et al. 2020). Fear of violence and theft in shelters, fear of being returned to unsafe home environments, and lack of youth-specific and gender-identity-affirming services contribute to RHY being harder to engage and enumerate than adult homeless (Ha et al. 2015; Pedersen et al. 2016; Shelton 2015). Nevertheless, it is estimated that over 3 million youth experience an episode of homelessness each year (Morton et al. 2018).

Runaway and homeless youth (including both minors and youth aged 18 and over) are regularly solicited with opportunities for paid sex, in person and online. Often, youth are targeted for sex specifically because of their use of homeless services, making them among the most vulnerable for human trafficking in the United States (Fogel et al. 2017; U.S. Department of State 2019). Other studies suggest that LGBTQ youth experiencing homelessness have disproportionately high rates of engaging in the sex trade to survive (Kattari and Begun 2017; National Human Trafficking Hotline 2020). However, the underground nature of sex and labor trafficking makes victim identification problematic for researchers, victim services, and law enforcement. One problem is the lack of standardized measurement tools and methods to identify exploitative experiences as discrete incidents of trafficking. Inconsistency in the definition of trafficking at the state and local level is also problematic. Additionally, the prioritization of research on sex over labor trafficking and the conflation of sex work/sex trafficking terminology confuse our ability to understand all forms of trafficking. Altogether, these problems mean that accurate prevalence estimates of trafficked RHY in the United States are not available (Dank et al. 2017; Fedina and DeForge 2017; Gibbons et al. 2020; Nawyn et al. 2013; Walts 2017; Weiner and Hala 2008).

*1.2. Prior Estimates of the Prevalence of Trafficking among RHY*

Several studies of the exploitative sex and labor situations experienced by RHY have been carried out in an effort to elucidate the issue. Table 1 summarizes the existing studies and highlights the key characteristics of the samples studied and the estimated prevalence of sex, labor, and any trafficking (when reported). We searched our university library databases and Google Scholar to locate trafficking prevalence estimates among RHY. Title and topic searches were performed for the search terms "trafficking" and "youth" and only articles using a sample of RHY were used. This is not a formal systematic review; rather, we present these studies details in order to better describe the preliminary nature of this limited literature. While this growing body of work on sex and labor trafficking among RHY has been critical for raising public awareness and resources, much of this work is not published in the peer-reviewed literature, and the methods utilized vary considerably. The majority of existing studies identified young people who had presented themselves to shelters, drop-in centers, and/or programs operated by agencies serving youth experiencing homeless, which can overlook young people who have managed to avoid or access the system intermittently. The majority of the studies have also been conducted in major northeastern and/or southwestern urban centers, neglecting the southern United States. Finally, the survey instruments have not been systematically evaluated and reflect a strong preferential primary focus on commercial sexual exploitation rather than all forms of trafficking.

The first critical study was a groundbreaking study of young people experiencing homelessness at Covenant House New York, a major youth-dedicated homeless shelter and service center. Researchers there found, in a sample of 174 youth, almost 23% had a trafficking experience based on force, fraud, or coercion (FFC) or engaged in survival sex (trading sex for basic necessities such as food or shelter due to no other options) while homeless. Close to 3% of the youth in the sample reported forced labor servitude. LGBTQ youth were overrepresented among the trafficked, with all four transgender youth reporting survival sex as their only option for income (Bigelsen and Vuotto 2013).

This study has been replicated several times using Covenant House youth in different cities. In 2015, a sample of 99 youth at Covenant House New Orleans revealed that 14% were involved in some form of human trafficking, including 11% in sex trafficking, 5% in labor trafficking, and two respondents in both. Fifteen percent of respondents participated in survival sex, with young women being more likely to report being paid for sex (Murphy et al. 2015). Another study of Covenant House youth in ten different cities found that among 641 respondents, 19% were involved in some form of human trafficking, including over 14% in sex trafficking, 8% in labor trafficking, and 3% in both. Young women, LGBTQ youth, young people with a history of foster care, as well as Latinx and American Indian/Native Canadian/Alaska Native young people were overrepresented among trafficking victims (Murphy 2016). Murphy describes a wide variety of ways young people were initiated into exploitative situations, sometimes with deception and other times having some awareness of the nature of the work situations. Sex trafficking experiences included modeling, pornography, and escort work, while labor trafficking included selling drugs, stolen goods, magazines, and pyramid schemes, among others (Murphy 2018).

An additional sample of 270 Covenant House youth in three additional cities was collected between 2016 and 2017 (Schilling Wolfe et al. 2018). In this study, researchers found that 20% were involved in some form of human trafficking, including almost 17% in sex trafficking, almost 3% in labor trafficking, and 2.6% in both; women, transgender, bisexual, and Latinx respondents were overrepresented among trafficking victims. Finally, in an effort to develop and validate a new screening tool, a separate study of 307 Covenant House residents in New Jersey found that 9.5% had a trafficking experience, including 6.5% in sex trafficking, 4.5% in labor trafficking, and 1.6% of them in both (Chisolm-Straker et al. 2019).

**Table 1.** Chronological summary of existing studies of trafficking among homeless youth, 2013–2019.

| Author(s) (Year) | Study Type | Sample Size | Age Range/Mean SD | Geographic Location(s) | Data Collection Venues | Instrument(s) Used | Estimated Prevalence Sex Trafficking | Estimated Prevalence Labor Trafficking | Estimated Prevalence Any Trafficking |
|---|---|---|---|---|---|---|---|---|---|
| Bigelsen and Vuotto (2013) | Agency Report | N = 174 | Range = 18–23 | New York City, NY | Covenant House Shelters and Outreach Van | Human Trafficking Interview and Assessment Measure (HTIAM-14) | 12% | 2.9% | 23% |
| Roe-Sepowitz et al. (2014) | Agency Report | N = 246 | Range = 18–25 M = 21.3 SD = 1.96 | Phoenix and Tucson, AZ | Trans. housing, drop-in centers, and on the streets | HTIAM-14; Arizona State University Office of Sex Trafficking Intervention Research Survey | 25.6% | N/A | 25.6% |
| Gibbs et al. (2015) | Peer-Reviewed Published Study | N = 111 | Not Reported | New York City, NY | Homeless youth-serving agency | Interviews and case narratives | 29% | N/A | 29% |
| Murphy et al. (2015) | Agency Report | N = 99 | Range = 18–23 M = 20.5 SD = 1.4 | New Orleans, LA | Covenant House Shelter | HTIAM-14 | 11% | 5% | 14% |
| Roe-Sepowitz et al. (2016) | Agency Report | N = 215 | Range = 18–25 M = 21.2 SD = 2.26 | Phoenix and Tucson, Arizona | Trans. housing, drop-in centers, and on the streets | Youth Experiences Survey (YES) | 35.8% | N/A | 35.8% |
| Murphy (2016) | Agency Report | N = 641 | Range = 17–25 | Anchorage, AK; Atlanta, GA; Detroit, MI; Fort Lauderdale, FL; Los Angeles, CA; Oakland, CA; St. Louis, MO; Toronto, ON; Vancouver, BC | Covenant House Shelters | HTIAM-14 | 14% | 8.1% | 19% |
| Roe-Sepowitz et al. (2016) | Agency Report | N = 199 | Range = 18–25 M = 21.1 SD = 2.17 | Phoenix and Tucson, Arizona | Trans. housing, drop-in centers, and on the streets | YES | 33.2% | N/A | 33.2% |
| Roe-Sepowitz et al. (2017) | Agency Report | N = 187 | Range = 18–25 M = 21.1 SD = 2.23 | Phoenix and Tucson, AZ | Trans. housing, drop-in centers, and on the streets | YES | 31% | 32.1% | 42.8% |
| Middleton et al. (2018) | Peer-Reviewed Published Study | N = 132 | Range = 12–25 M = 19.65 SD = 3.73 | Louisville, KY and Southern Indiana | 8 homeless youth-serving agencies | YES | 41.2% | N/A | 41.2% |

**Table 1.** *Cont.*

| Author(s) (Year) | Study Type | Sample Size | Age Range/Mean SD | Geographic Location(s) | Data Collection Venues | Instrument(s) Used | Estimated Prevalence Sex Trafficking | Estimated Prevalence Labor Trafficking | Estimated Prevalence Any Trafficking |
|---|---|---|---|---|---|---|---|---|---|
| Schilling Wolfe et al. (2018) | Agency Report | N = 270 | M = 20.7 SD = 2.0 | Philadelphia, PA Phoenix, AZ Washington, DC | 5 shelters and youth service orgs. | HTIAM-10 | 17% | 6% | 20% |
| Roe-Sepowitz et al. (2018) | Agency Report | N = 179 | Range = 18–25 M = 20.8 SD = 2.20 | Phoenix and Tucson, AZ | Trans. housing, drop-in centers, and on the streets | YES | 34.6% | 30.7% | 44.7% |
| Chisolm-Straker et al. (2019) | Peer-Reviewed Published Study | N = 307 | Range = 18–22 M = 19.5 | New Jersey | Covenant House Shelters | HTIAM-14; Quick Youth Indicators for Trafficking (QYIT) | 6.5% | 4.6% | 9.5% |
| Greeson et al. (2019) | Peer-Reviewed Published Study | N = 270 | M = 20.7 SD = 2.0 | Philadelphia, PA Phoenix, AZ Washington, DC | Homeless youth-serving agencies | HTIAM-10 | 17% | N/A | 17% |
| Mostajabian et al. (2019) | Peer-Reviewed Published Study | N = 120 | Range = 18–21 M = 19.0 SD = 1.0 | Houston, TX | Youth shelter | Human Trafficking Screening Tool (HTST) | 25.8% | 54.2% | N/A |
| Roe-Sepowitz et al. (2019) | Agency Report | N = 167 | Range = 18–25 M = 20.9 SD = 2.34 | Phoenix and Tucson, AZ | Trans. housing, drop-in centers, and on the streets | YES | 38.9% | 43.1% | 53.3% |
| Roe-Sepowitz et al. (2020) | Agency Report | N = 81 | Range = 18–25 M = 20.6 SD = N/A | Phoenix and Tucson, AZ | Trans. housing, drop-in centers, and on the streets | YES | 29.6% | 24.7% | 45.7% |

In a separate study of minors collected over a 2.5-year period at three different sites, one of which exclusively worked with RHY, researchers found that 55% of 201 minors reported sex trafficking experiences and 5% both sex and labor. Among those engaged by the agency specifically serving homeless youth, 29% reported trafficking experiences, mostly in the form of survival sex. The overall sample reported having someone who facilitated and controlled their sexual transactions, but 78% of trafficked youth experiencing homelessness reported facilitating their own sex trades for a place to stay rather than money (Gibbs et al. 2015).

In addition to the series of Covenant House studies, a longitudinal study in Arizona utilized the Youth Experiences Survey (YES) tool to estimate the prevalence of sex trafficking among young people experiencing homelessness in Phoenix and Tucson, Arizona. Data were collected each year from 2014–2020, and the seven-year average for sex trafficking experiences was 32.7%, with LGBTQ youth being at higher risk for reporting being trafficked. The survey instrument also asked about technology and smart phones, giving valuable information about how young people are recruited and/or surveilled for trafficking (Roe-Sepowitz et al. 2016). Researchers in counties located in both Kentucky and Indiana used the YES to assess sex trafficking among 132 youth experiencing homelessness in the fall of 2016 (Middleton et al. 2018). Over 41% of these people reported sex trafficking experiences, 47.6% of whom were women. Two out of the three respondents who identified as "other gender" also reported sex trafficking. Of those who reported sex trafficking, 64.8% were heterosexual and 27.8% identified as LGBTQ (Middleton et al. 2018).

More recently, researchers in Philadelphia, Washington, D.C., and Phoenix were interested not only in prevalence but differences in who was considered trafficked due to FFC versus due to being underage. Seventeen percent of the 270 respondents had been trafficked and were disproportionately female, bisexual, and Latinx. Those who were victimized by FFC were also more likely to be female, less educated, and bi or pansexual (Greeson et al. 2019). Finally, as part of a larger study to test the HTST, Mostajabian et al. (2019) found that 25.8% of RHY in a large shelter in Houston, Texas had been victims of CSE and 54.1% victims of labor trafficking.

In this study, we endeavor to begin to address the three identified gaps by (1) collecting data across a wider range of community settings, including RHY in shelters, on the streets, in motels/hotels, and in other public locations; (2) focusing on the metro-Atlanta area, a broad geographic area located in the southeastern U.S.; (3) utilizing a well-established screening tool designed to measure commercial sexual exploitation as well as other forms of labor trafficking.

## 2. Methods

Data from the 2018 Atlanta Youth Count (AYC18) are used in this study. The AYC18 was a cross-sectional study designed to estimate the size and describe the population of youth experiencing homelessness. The geographic focus was metro-Atlanta, which extended beyond the formal city limits and included five adjacent counties that comprise the metropolitan area, including Fulton, DeKalb, Clayton, Cobb, and Gwinnett counties. The central research question guiding this paper is: what is the prevalence of sex and labor trafficking among RHY in metro-Atlanta?

### 2.1. Research Design

Informed by over a year of ethnographic preparatory field work, the research team conducted a cross-sectional survey of RHY. The survey instrument included a range of questions about respondents' demographic and social background, the time and nature of their experiences with homelessness, contact with various service systems, health status, and trafficking. To be eligible to participate, young people had to state that they: (1) were between 14–25 years of age; (2) had not had a permanent stable residence of their own for at least one day in the prior 30 days; and, (3) were living independently without consistent parental or family support. The young people were offered the choice of completing the

survey themselves or having one of the student researchers administer the questionnaire in an interview format.

In all aspects, the research strove to protect the confidentiality of the survey data, and no information was made available to service providers present on site or in the field. If a young person requested specific help, they were directed to the service providers. Everyone we encountered was offered a booklet describing the services available to them in the area, and every respondent who completed the survey received an untraceable USD 10 gift card. All procedures used were reviewed and approved by the first author's Institutional Review Board, prior to initiating the study.

### 2.2. Sample and Data Collection

Student researchers and project staff conducted a year of ethnographic fieldwork prior to initiating survey data collection. During this ethnographic period, we developed relationships with homeless service providers and youth we encountered. This period ensured trust within the community and helped the research team identify potential locations for recruiting youth and plan the survey data collection phase. This pre-field work also facilitated the training of the student researchers on how to approach and interact with potential respondents and to navigate the varied community settings. We utilized capture–recapture methods to identify specific community locations from which to recruit the sample. This step involved identifying time periods and locations most likely to be successful in recruiting RHY in shelters, street and community venues, and other service locations identified during the ethnographic phase (Hay and McKeganey 1996; Mastro et al. 1994; Otis et al. 1978; Roberts and Brewer 2006). Data were collected in three 10-day sweeps of the region spread over three months to collect systematic samples of RHY designed to construct more rigorous estimates of the population as well as the prevalence of labor and sex trafficking. Capture–recapture methods rely not only on the collection of systematic information across a defined geographic area, but also on the frequency of repeated contacts to build estimates of the underlying invisible population (Hay and McKeganey 1996; Mastro et al. 1994; Otis et al. 1978; Roberts and Brewer 2006).

To ensure that the youth surveyed could not be identified or traced, the survey was anonymous; however, the survey did include a discrete series of questions that the research team used to construct a unique identifier that was used identify young people the researchers believed participated more than once.

Small teams, comprised of service providers and student research assistants, were assigned to recruit, screen, and administer surveys to all eligible young people encountered in the time-sampled locations. Locations were determined by the previous fieldwork and relied heavily on service providers' changing knowledge of the youth' activities. We strategically selected areas and times so that estimates of the population could be ascertained after the study. Student researchers, who were usually in the same age range as the potential respondents, were responsible for recruiting potential subjects and conducting the interviews. Service providers assisted in identifying potential participants they knew were eligible for the study and referring them to the research team. Additionally, student researchers were trained in recognizing potential youth by asking community members about their knowledge of young people who may be experiencing "housing difficulties." Due to the survey instrument's sensitive nature, eligible participants were given the option to either complete the survey on their own or with an interviewer. Youth completed surveys in service provider locations along with public areas where they were known to congregate. In total, we contacted 764 while we were in the field. After extensive cleaning and removing duplicate surveys, our study yielded a sample of 564 youth. A more extensive, publicly available overview of the methods used in collecting these data is available from the first author.

*2.3. Variables*

Experiences with human trafficking was measured using a slightly modified version of the Human Trafficking Screening Tool or HTST (Dank et al. 2017), a 17-item scale developed for this target population designed to identify a range of labor and sex trafficking experiences in four broad categories: force, fraud, coercion, and commercial sexual exploitation (CSE). The instrument begins with a broad introduction and question about the ways they made money, specifically calling attention to the types of work youth on the streets do, and then asks them about whether or not these work experiences involved four general types of trafficking.

The original questionnaire asked whether each of the 17 experiences had ever happened and if they had happened within the last year. We supplemented the questionnaire by also asking whether each experience had happened "while you were homeless." For this paper, we focus on the sub-sample of youth who responded affirmatively to at least one of these questions (n = 519). We constructed a series of binary variables to indicate whether the experience had occurred within the three time periods (ever, last year, while homeless). We also computed summary binary variables to indicate if the youth had experienced one or more of the four types (i.e., force, fraud, coercion, or CSE) or any form of trafficking.

We examine prevalence of each type of trafficking by various demographic constructs, we used the young person's reported age (in years) as a series of categorical variables. We coded the participants' gender identity using a series of dummy variables (cisgender women = 1; transgender or other gender identity = 1; v. cisgender men = 0) in order to examine the experiences of cisgender women and transgender/gender nonconforming individuals separately from cisgender men. We coded the respondents' sexuality identity as a dummy variable as well (lesbian, gay, or bisexual/LGB = 1; straight or heterosexual = 0). We utilized two binary dummy variables to identify respondents' race/ethnicity (Black or African American = 1; other = 0) and overall education level (completed high school or a GED = 1; not having a high school diploma = 0). Finally, we also used dummy variables to indicate if the current episode was their first time experiencing homeless (1 = first time homeless; 0 = not their first time) and whether they had been homeless for longer than one year (1 = homeless for more than a year; 0 = homeless for less than 365 days).

*2.4. Analysis*

The data analysis consisted of three major components. First, we computed summary statistics to describe the overall sample, focusing on the demographic and background variables included in the multivariate analyses. Second, we examined the overall percentages of affirmative responses to each of the HTST items by trafficking category as well as any form of trafficking. Finally, to better understand and evaluate whether or not particular youth are more or less vulnerable to the various forms of trafficking, we used logistic regression analysis to examine trafficking variation across the core demographic and social variables.

## 3. Results

Table 2 provides an examination of the RHY in Atlanta in 2018. The mean age of the sample was 21.43 (SD = 2.35). The majority of the participants were either Black or African American (55.9%) or multi-racial (32.4%), with the majority of this latter group reporting being Black or African American and some other racial or ethnic group. While the majority reported being cisgender men (59.7%) or cisgender women (33.9%), we did collect data from 33 (6.5%) individuals who identified as transgender or some other gender identity. With regard to sexuality, 27.5% of the sample described their sexual identity as being lesbian, gay, or bisexual, with the remainder (72.5%) saying that they were "straight" or heterosexual. Two-thirds (66%) had completed high school or their GED. We also asked about their prior involvement with child welfare, state mandated mental health treatment, juvenile legal system, and foster care. Over half (56.2%) of the sample reported having contact with one or more of these systems prior to the age of 18. In terms of their experiences with homelessness, 198 (37.1%) told us that this was their first time experiencing homelessness,

with 97 or 18.4% indicating that the duration of their current episode of homelessness was less than 1 month.

**Table 2.** Demographic profile of runaway and homeless youth (RHY) participants, 2018 Atlanta Youth Count (n = 564).

| Demographic Characteristic | n | % |
|---|---|---|
| Age (Mean = 21.43; SD = 2.35) | | |
| 17 or Younger | 13 | 2.3 |
| 18–19 Years Old | 127 | 22.5 |
| 20–21 Years Old | 166 | 29.4 |
| 22–23 Years Old | 111 | 19.7 |
| 24–25 Years Old | 147 | 26.1 |
| Race/Ethnicity | | |
| Black; African American | 300 | 55.9 |
| Multiracial | 174 | 32.4 |
| White; Caucasian | 40 | 7.4 |
| Other Race; Ethnicity | 23 | 4.3 |
| Gender Identity | | |
| Cisgender Man | 305 | 59.7 |
| Cisgender Woman | 173 | 33.9 |
| Transgender or Other Gender Identity | 33 | 6.5 |
| Sexuality Identity | | |
| Straight (Heterosexual) | 387 | 72.5 |
| Lesbian, Gay, or Bisexual (LGB) | 147 | 27.5 |
| Education (Hs Diploma or Better) | 372 | 66.0 |
| System Involvement as Child (Before Age 18) | 317 | 56.2 |
| First Time Homeless (Yes) | 198 | 37.1 |
| Time Homeless (Current Episode) | | |
| Less Than 1 Month/30 Days | 97 | 18.4 |
| 1–2 Months (30–89 Days) | 95 | 18.0 |
| 3–6 Months (90–180 Days) | 115 | 21.8 |
| 6 Months–1 Year (181–364 Days) | 76 | 14.4 |
| 1 Year or More (More Than 365 Days) | 145 | 27.5 |

A total of 519 of the youth who responded completed the HTST (92%). Table 3 summarizes the percentage of respondents who indicated they had experienced the 17 types of trafficking included in the questionnaire, ever, in the last year, and while they were homeless. In terms of the most frequent, nearly one out of three (31.6%) said someone had refused to pay them or paid them less than they were promised, while nearly one out of four of the young people reported that someone had kept all or most of their money or pay (22.4%) or physically forced them to do something that they did not feel comfortable doing (22.2%). A comparison of the first column with the latter two columns suggests that a larger percentage of these young people's trafficking experiences occurred either within the last year and/or while they were homeless. The remarkable similarity in the reported prevalence of trafficking in the last year and while they were homeless, regardless of type, suggests that these forms of abuse could be both a consequence and possibly a force that preceded their becoming homeless.

Table 4 classifies the item responses based on the four types of trafficking as well as the overall prevalence of trafficking. Coercion (40.3%) and fraud (39.5%) stand out as the most common types, regardless of whether it is measured over the course of their lives or more recently. Interestingly, CSE was the least common form, with 19.5% of youth reporting having been sexually exploited in their lifetimes and 16.2% and 15.6% in the last year and/or while they were homeless. Taken together, over half (56.6%) of the youth reported experiencing any form of trafficking in their lifetimes, with the majority of it appearing

to have occurred within the last year (45.3%) or while they were homeless (39.9%). This pattern of results indicates that homelessness clearly increases young people's risk for being trafficked regardless of the form it takes. Because the HTST is a screening tool, we must underscore that a youth could have had a single episode or multiple events in mind when they answered "yes" to any of these items. Thus, we are unable to distinguish or estimate the number of discrete trafficking events only that a particular type of trafficking occurred.

**Table 3.** Number and percentage of youth who responded affirmatively to the Human Trafficking Screening Tool (HTST), 2018 Atlanta Youth Count (n = 519).

| Category | Item | Ever | Last Year | While Homeless |
|---|---|---|---|---|
| | *Did someone you work for …* | % | % | % |
| Force | Physically force you to do something you didn't feel comfortable doing? | 22.2 | 11.2 | 11.2 |
| | Lock you up, restrain you, or prevent you from leaving? | 16.4 | 9.6 | 9.1 |
| | Physically harm you in any way (beat, slap, hit, kick, punch or burn)? | 20.8 | 11.4 | 10.8 |
| Fraud | Trick you into doing different work than was promised? | 16.2 | 14.6 | 15.2 |
| | Make you sign a document without understanding what it stated, like a work contract? | 15.2 | 9.1 | 8.7 |
| | Refuse to pay you or pay you less than they promised? | 31.6 | 22.0 | 19.5 |
| Coercion | Restrict or control where you went or who you talked to? | 21.4 | 13.9 | 13.7 |
| | Deprive you of sleep, food, water or medical care? | 16.4 | 11.2 | 10.8 |
| | Not let you contact family or friends even when you weren't working? | 14.5 | 10.2 | 8.3 |
| | Keep all or most of your money or pay? | 22.4 | 14.6 | 13.9 |
| | Keep your ID documents (e.g., ID card, license, passport, social security card, birth certificate) from you? | 16.2 | 9.8 | 10.6 |
| | Threaten to get you deported? | 7.7 | 4.6 | 5.0 |
| | Threaten to harm you or your family or pet? | 16.0 | 10.0 | 9.8 |
| | Physically harm or threaten a co-worker or friend? | 13.9 | 9.4 | 7.7 |
| Commercial Sexual Exploitation (CSE) | Force you to do something sexually that you didn't feel comfortable doing? | 16.8 | 11.4 | 11.8 |
| | Put your photo on the internet to find clients to trade sex with? | 11.0 | 7.7 | 7.5 |
| | Force you to engage in sexual acts with family, friends, or business associates for money or favors? | 10.8 | 7.9 | 7.3 |
| | Force you to trade sex for money, shelter, food or anything else through online websites, escort services, street prostitution, informal arrangements brothels, fake massage businesses, or strip clubs? | 12.9 | 9.8 | 9.4 |

**Table 4.** Number and percentage of youth who reported each of the four types and any type of trafficking assessed in the human trafficking screening tool (HTST), 2018 Atlanta Youth Count (n = 519).

| Category of Trafficking | Ever | Last Year | While Homeless |
|---|---|---|---|
| | % | % | % |
| Force | 31.4 | 19.1 | 17.9 |
| Fraud | 39.5 | 30.8 | 25.2 |
| Coercion | 40.3 | 31.6 | 29.3 |
| Commercial Sexual Exploitation (cse) | 19.5 | 16.2 | 15.6 |
| Any Trafficking | 56.6 | 45.3 | 39.9 |

Table 5 presents the logistic regression models for the four categories of trafficking and any trafficking that occurred over the young person's lifetime. Youth who had prior contact with any of the formal child-serving systems—child welfare, state mandated mental health, juvenile legal system, and/or foster care—are twice as likely (O.R. = 2.001, $p \leq 0.001$) to report having experienced any type of trafficking, regardless of the type or category (O.R.s

range from 1.339 to 1.849 across the four types and are significant at the $p \leq 0.01$ level). Cisgender women also have a much higher likelihood of experiencing CSE (O.R. = 4.424, $p \leq 0.001$) as do transgender individuals (O.R. = 7.691, $p \leq 0.001$). Transgender and sexual minority young people also report a greater likelihood of being forced or physically harmed in some way (O.R.s = 3.197 and 1.937, $p \leq 0.01$, respectively).

**Table 5.** Logistic regression analyses of each type of and any trafficking ever experienced among youth experiencing homelessness, 2018 Atlanta Youth Count (n = 455).

| Predictor | Force | Fraud | Coercion | CSE | Any Trafficking |
|---|---|---|---|---|---|
| Age (in Years) | 0.978 | 1.043 | 0.924 + | 0.909 | 0.983 |
| Race (Black) | 0.660 | 0.723 | 0.758 | 1.063 | 0.699 |
| Race (Multiracial) | 0.756 | 0.713 | 0.917 | 1.155 | 0.754 |
| Gender (Cisgender Woman) | 1.408 | 1.152 | 1.200 | 4.424 *** | 1.147 |
| Transgender | 3.197 ** | 1.910 | 1.661 | 7.691 *** | 1.545 |
| Sexual Minority (LGB) | 1.937 ** | 1.135 | 1.083 | 1.097 | 1.226 |
| Education (HS or better) | 1.143 | 0.685 + | 1.042 | 1.285 | 0.938 |
| System Involved | 1.823 ** | 1.849 ** | 1.805 ** | 1.339 ** | 2.001 *** |
| First Time Homeless | 1.153 | 0.984 | 0.759 | 0.658 | 0.902 |
| Homeless for More Than 1 Year | 1.138 | 1.212 | 1.468 + | 1.763 + | 1.473 + |
| −2 Log Likelihood | 515.255 | 585.203 | 586.230 | 354.667 | 602.394 |
| Overall $X^2$ | 34.571 *** | 22.953 * | 21.014 * | 55.880 *** | 23.507 ** |
| Nagelkerke $R^2$ | 0.104 | 0.067 | 0.061 | 0.194 | 0.067 |

Note: + $p \leq 0.10$, * $p \leq 0.05$, ** $p \leq 0.01$, *** $p \leq 0.001$.

In Table 6, we present the parallel models for trafficking experiences that occurred while the respondents were homeless. Because the pattern of responses for trafficking while homeless are nearly identical to those for trafficking in the last year, we only present those for trafficking while homeless (the latter results are available upon request from the first author). Overall, the explained variance in these models is much stronger and the patterns of results very clear and similar to lifetime trafficking experiences. The most consistent risk factor for experiencing any trafficking and all four trafficking categories is having prior system involvement (O.R.s range from 2.385 to 1.733 and all are significant at the $p \leq 0.05$ or better). Transgender youth similarly stand out for being at risk of experiencing all four types and any form of trafficking while they are homeless, particularly physical force (O.R. = 5.008, $p \leq 0.001$) and sex trafficking (O.R. = 8.961, $p \leq 0.001$). Cisgender women are significantly more likely to experience physical force (O.R. = 2.157, $p \leq 0.01$) or CSE (3.462, $p \leq 0.001$) while they are on the streets or other homeless circumstances.

**Table 6.** Logistic Regression analyses of each type of and any trafficking experienced while homeless among youth experiencing homelessness, 2018 Atlanta Youth Count (n = 455).

| Predictor | Force | Fraud | Coercion | CSE | Any Trafficking |
|---|---|---|---|---|---|
| Age (in Years) | 0.951 | 0.993 | 0.961 | 0.919 | 0.967 |
| Race (Black) | 0.715 | 0.600 | 0.838 | 1.510 | 0.854 |
| Race (Multiracial) | 0.660 | 0.567 | 0.886 | 1.422 | 0.868 |
| Gender (Cisgender Woman) | 2.157 ** | 1.243 | 1.067 | 3.462 *** | 1.370 |
| Transgender | 5.008 *** | 3.671 ** | 2.475 * | 8.961 *** | 3.51 8** |
| Sexual Minority (LGB) | 1.252 | 1.328 | 1.317 | 1.358 | 1.277 |
| Education (HS or better) | 1.143 | 1.148 | 0.782 | 1.225 | 0.987 |
| System Involved | 1.834 * | 1.733 * | 1.893 ** | 2.385 ** | 2.001 *** |
| First Time Homeless | 0.825 | 0.820 | 1.007 | 0.633 | 0.814 |
| Homeless for More Than 1 Year | 1.317 | 1.797 * | 2.032 ** | 1.963 * | 2.112 *** |
| −2 Log Likelihood | 380.215 | 473.032 | 507.469 | 310.245 | 574.184 |
| Overall $X^2$ | 30.332 *** | 30.287 *** | 29.453 *** | 48.334 *** | 32.129 *** |
| Nagelkerke $R^2$ | 0.109 | 0.096 | 0.090 | 0.185 | 0.093 |

Note: * $p \leq 0.05$, ** $p \leq 0.01$, *** $p \leq 0.001$.

These models highlight an additional risk factor: being homeless for longer periods of time. Specifically, when young people have been homeless for more than a year, young people reported significantly higher odds of experiencing some type of fraud (O.R. = 1.797; $p \leq 0.05$), coercion (O.R. = 2.032, $p \leq 0.01$), and commercial sexual exploitation (O.R. = 1.963, $p \leq 0.05$) as well as being twice as likely to experience any form of trafficking (O.R. = 2.112, $p \leq 0.001$).

## 4. Discussion

Our findings suggest that trafficking is a remarkably common occurrence among young people experiencing homeless in the metro-Atlanta area, with approximately half of the homeless youth we encountered reporting some form of exploitation in their lifetime and approximately four out of 10 participants indicating they had been exploited immediately before or while they were homeless. While our estimates are generally similar with other studies that call attention to the problem of trafficking among RHY (Countryman-Roswurm and Bolin 2014; Dank et al. 2015; Gibbs et al. 2018; Middleton et al. 2018; Dank et al. 2017), the occurrence of trafficking is on parr or slightly higher than studies conducted in other parts of the country, offering some support of claims that metro-Atlanta is a major hub for human trafficking in the United States (Bailey and Wade 2014; Ellis 2020; Parker 2018). The extraordinary levels of income inequality in the city (Bloomberg.Com 2018; Berube 2018), stratification in the local labor market (Kligman and Limoncelli 2005), and being a known as the "Black gay mecca" of the South (Bartone 2015; Marini 2018; Schwarz and Britton 2015) may contribute both to the large numbers of runaway and homeless young people and the number of youth who are trafficked.

At the same time, our research underscores that labor trafficking is perhaps a more significant form of exploitation than commercial sexual exploitation. Our data suggest that the rates for various forms of labor trafficking—whether in their lifetime, the last year, or while homeless—are always higher and, in some cases, twice as high as commercial sexual exploitation, with coercion and fraud being the most prevalent forms of exploitation. It is important to remember, though, that the HTST focuses on abuse occurring in informal work settings (e.g., selling drugs) and may not capture mistreatment in more formal work settings. Because the HTST focuses on measuring the simple occurrence of various types of exploitation during work, as opposed to describing discrete episodes or periods of abuse, it suggests a strong pattern of co-occurrence in the various forms of exploitation in the lived experiences of RHY. This finding is consistent with other studies that describe the multiple vulnerabilities of RHY (Bender et al. 2015; Dank et al. 2015). Our study adds to this growing body of research and underscores the need to more deeply understand the nature of these connections as well as how, when, and the patterns of various forms of abuse.

In the case of RHY, we believe it is particularly noteworthy that both types of trafficking appeared to have occurred before the young person reported they were homeless, sometimes even shortly before their current episode of homelessness. This begs the question of the role trafficking plays in leading young people to become homeless. Prior research, as well our own findings, highlights the greater vulnerabilities of system-involved youth for abuse and exploitation and suggests that many of these young people may end up homeless in their efforts to escape problematic situations (Bender et al. 2015; Dworsky et al. 2013; Forge et al. 2018; Ream and Forge 2014; Zlotnick et al. 2012). While the rates of reported trafficking are somewhat lower in our sample for young people while they are homeless, approximately one out of four reported experiencing coercion or fraud and nearly one out of five indicated being forced or commercially sexually exploited. In short, this pattern of responses highlights that youth experiencing homelessness are exceptionally vulnerable, and more important, trafficking may be both a cause and a consequence of their experiencing homelessness.

At the most general level, our multivariate analyses reinforce prior studies that have sought to identify risk factors for trafficking among young people. Similar to others, we

found that young cisgender women, transgender, and sexual minority youth are at greater risk of experiencing trafficking (Gibbs et al. 2018; Greeson et al. 2019; Murphy 2016; Schilling Wolfe et al. 2018). In line with other studies (Bender et al. 2015; Forge et al. 2018; Gibbs et al. 2018; Dank et al. 2017), our results underscore the particular vulnerabilities of youth who are currently or have been involved in child-serving systems such as foster care, child welfare, or juvenile legal system. Indeed, in our study, system involvement stands out as the strongest and most consistent risk factor for experiencing trafficking among RHY.

Our study has two important limitations. First, our study is limited in its restricted geographic focus to the Atlanta metropolitan area. Given the elusive and hidden nature of the population, it would be very difficult to get a truly nationally or internationally representative sample of RHY. Nevertheless, as noted above, our targeted focus on metro-Atlanta provides new insights on an urban area that has not been systematically studied in the existing literature. In addition, our understanding of trafficking in these young people's lives is limited because of our reliance on the HTST. While it is an excellent screening instrument and useful for establishing aggregate rates and the need for trafficking-related services in this population, it does not facilitate an in-depth understanding of the nature of these young people's trafficking experiences. Indeed, Dank et al. (2017) recommend that young people who are screened using the HTST be interviewed by trained clinicians to more fully understand the nature of their experiences and the extent of the trauma they have endured. In practice, as a research instrument, it is important to note that a single instance of trafficking could have been in mind when a young person checked "yes" to multiple questions. Informal conversations and anecdotes we heard from the young people we encountered in the field underscore this complex reality. We heard from multiple young people who reported being trafficked that their experiences involved much more than discrete episodes of exploitation. They often unfolded in the context of complex and abusive relationships that sometimes extended over significant periods of time.

## 5. Conclusions

Our study adds to the growing body of research that documents high rates of labor and sex trafficking among RHY. When compared with other studies conducted in other parts of the U.S., though, RHY in metro-Atlanta appear to be especially vulnerable to trafficking. Similar to other studies, we found that young women, youth who identify as transgender, and especially young people with prior system involvement are exceptionally vulnerable to experiencing trafficking. The increased risk among youth who have had prior system involvement, however, also points to a potentially important prevention opportunity: expanding and strengthening the support systems for young people who are preparing to leave child welfare, foster care, and juvenile justice programs. Indeed, we believe that increasing the array of services and financial supports to young people leaving various child-serving systems would reduce the number of youth who experience homelessness and trafficking and, in turn, help them make a more successful transition to adulthood. Our findings and field experiences further underscore the complex nature of trafficking and the need for more in-depth research to better understand the causal pathways that result in young people experiencing trafficking. A deeper understanding of these complex processes would contribute to a better scientific and public understanding of why various forms of trafficking persist and what factors contribute to young people's abilities to escape this form of abuse.

**Author Contributions:** Conceptualization: E.R.W., R.S., N.F., and E.R.; methodology: E.R.W., A.L., N.F., and E.R.; data analysis: E.R.W. and A.L.; data collection, cleaning, and coding: A.L., N.F., K.T., M.H., Z.W., M.T.-H., A.D., and C.W.; writing—original draft preparation: E.R.W., K.T., and A.L. All authors have read and agreed to the published version of the manuscript.

**Funding:** This project was supported by Award No. 2016-MU-MU-0002 (Eric R. Wright, PhD., Principal Investigator), awarded by the National Institute of Justice, Office of Justice Programs, U.S. Department of Justice. The opinions, findings, and conclusions or recommendations expressed in this publication are those of the author(s) and do not necessarily reflect those of the Department of Justice.

**Institutional Review Board Statement:** The study was conducted according to the guidelines of the Declaration of Helsinki and approved by the Institutional Review Board of Georgia State University (H18166, November 2017).

**Data Availability Statement:** The data presented in this study are available on request from the corresponding author. The data are not currently publicly available because the data deposit, required by the National Institute of Justice, is currently in process by the National Archive of Criminal Justice Data.

**Conflicts of Interest:** The authors declare no conflict of interest.

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
