# Peer review of "The Prevalence and Correlates of Labor and Sex Trafficking in a Community Sample of Youth Experiencing Homelessness in Metro-Atlanta"

_socsci, doi:10.3390/socsci10020032_

Round 1

Reviewer 1 Report

This is an excellent paper.  The study sample is larger than previous studies, and is likely more representative than previous work, based both on the heterogeneity of sample sites, and the three different periods from which samples were drawn.  The sample demographics are consistent with other large samples of RHY, although large samples focused on trafficking are less common.  The study used a standard instrument previously validated and which improves upon previous instruments.  The multivariate analyses were done appropriately (treating race/ethnicity as a single bivariate variable isn't ideal, but the sample was so strongly black/mixed and black, that I can appreciate it makes sense here).  The major findings both for ever and during homelessness are consistent across trafficking types, and strengthen the validity of the findings.  The robust findings for system involved youth are important, as are those for transgender youth trafficked while homeless.  The results are well interpreted in the discussion, with useful inferences about causality and bidirectionality.

A well done paper and a contribution to the field. 

Author Response

We thank the reviewer for their comments.  We have addressed the second reviewer's concerns in the other text field.

Reviewer 2 Report

In this paper Authors report estimates of commercial sexual exploitation, labour, and any trafficking collected during the Atlanta Youth Count 2018 field study (AYC18). Overall, an interesting paper but I see several areas of need for improving the manuscript:

In the Method Section, Authors should clearly indicate the central research question. What was the main purpose or goal of the study? This research has a quantitative character, so what was the study hypothesis? Of course, the authors have the right to not formulated a hypothesis, but they should justify it methodologically.

What was the Eligibility Criteria for systematic review in table 1? Did the authors use the Preferred Reporting Items for Systematic Reviews and Meta-Analyses (PRISMA) protocol or any other system? It should be clearly described.

Authors described sample but fail to describe sampling procedures. How were participants recruited?

Limitations of this study are missing.

Author Response

We thank the reviewer for their feedback. Below we list the concerns expressed and briefly describe in bold the changes we have made to the revised manuscript:

In the Method Section, Authors should clearly indicate the central research question. What was the main purpose or goal of the study? This research has a quantitative character, so what was the study hypothesis? Of course, the authors have the right to not formulated a hypothesis, but they should justify it methodologically.

We have expanded the introduction to the methods section to include a statement of the research question.  As an exploratory study, we did not include an hypothesis.

What was the Eligibility Criteria for systematic review in table 1? Did the authors use the Preferred Reporting Items for Systematic Reviews and Meta-Analyses (PRISMA) protocol or any other system? It should be clearly described.

This summary of the literature was not intended to be a systematic review or meta analysis.  Given the preliminary state of the literature, we included the table to highlight and guide our comparison of the existing studies, many of which were not formal peer-reviewed papers.  Based on the reviewer's comment, we have added additional language to clarify how we identified these studies and that they are not intended to be a PRISMA analysis.

Authors described sample but fail to describe sampling procedures. How were participants recruited?

We have added more details regarding the sampling procedures and how the participants were recruited in the field.

Limitations of this study are missing.

We have added a new paragraph discussing the limitations of the study.